# Investigation of Machining Performance of MQL and MQCL Hard Turning Using Nano Cutting Fluids

**Ngo Minh Tuan [1]**, **Tran Minh Duc [1]**, **Tran The Long [1]**, **Vu Lai Hoang [1]** and **Tran Bao Ngoc [2],***

1 Faculty of Mechanical Engineering, Thai Nguyen University of Technology, Thai Nguyen 250000, Vietnam; minhtuanngo@tnut.edu.vn (N.M.T.); minhduc@tnut.edu.vn (T.M.D.); tranthelong@tnut.edu.vn (T.T.L.); hoangvl@tnut.edu.vn (V.L.H.)
2 Department of Fluids Mechanic, Faculty of Automotive and Power Machinery Engineering, Thai Nguyen University of Technology, Thai Nguyen 250000, Vietnam
* Correspondence: baongoctran@tnut.edu.vn; Tel.: +84-917-191-356

**Abstract:** Cutting fluids used in the metal machining industry have exerted serious impacts on the environment and human health. In addition, the very high cutting heat and forces in machining-hardened steels have been a growing concern in the metal cutting field. Hence, new, eco-friendly cooling and lubricating techniques are necessary to study and develop. Minimum quantity lubrication (MQL) and minimum quantity cooling lubrication (MQCL) using nano cutting fluids have been proven as alternative solutions for machining difficult-to cut materials while retaining an environmentally friendly characteristic. Accordingly, this paper aims to analyze and evaluate the hard turning efficiency of 90CrSi ($60 \div 62$ HRC) steel using MQL and MQCL conditions, using $Al_2O_3$ and $MoS_2$ nano cutting fluids. The $2^{k-p}$ experimental design and analysis of variance (ANOVA) were used to study the influence of input parameters including fluid type, lubrication method, nanoparticle type, nanoparticle concentration, cutting speed and feed rate on surface roughness. The obtained results showed that the machinability of CNMG120404 TM T9125 carbide tools was improved and the highest machinable hardness was increased from 35 HRC to $60 \div 62$ HRC (rising by approximately $71.4 \div 77.1\%$) by using the nanofluid MQL and MQCL methods. Furthermore, MQCL gives better performance than MQL, and the $Al_2O_3$ nanofluid exhibits the better result in terms of surface roughness values than the $MoS_2$ nanofluid. Feed rate displays the strongest influence on surface roughness, while fluid type, nanoparticle concentration and cutting speed show low impacts. From these results, technical guidance will be provided for further studies using $Al_2O_3$ and $MoS_2$ nano cutting fluids for MQL and MQCL methods, as well as their application in machining practice.

**Keywords:** hard turning; hard machining; MQL; MQCL; nanoparticles; nano cutting fluid; difficult-to-cut material

## 1. Introduction

In recent years, machining difficult-to-cut materials has become a growing concern in the metal cutting field, in which hard turning has been widely used to improve dimensional accuracy, surface quality and machining productivity, as well as reduce the manufacturing cost and negative effects on the environment. This technology has received much attention and become an alternative to grinding processes due to its high productivity. In addition, it is suitable for complex profiles, reduces the use of cutting oil and yields good surface quality. However, very high friction in the cutting zone causes enormous heat, which accelerates the wear rate, reduces tool life and limits the cutting condition, as well as requiring high-quality cutting inserts [1]. One of the most effective ways to reduce the high cutting temperature and large cutting forces in hard machining processes is the application of a proper cooling lubricant technique. However, the introduction of the coolant into the cutting zone under flood conditions is not appropriate and can cause thermal shock, leading to tool breakage [2]. Hence, MQL and MQCL methods are commonly applied in hard

machining processes to overcome these problems and replace dry and flood conditions. For the MQL technique, a small amount of cutting fluid is directly sprayed into the contact zones in oil mist form to elicit a superior lubricating effect [3]. The influence of technological parameters in hard machining processes using MQL has been investigated in many studies. Most studies show that the MQL method gives better results in terms of surface quality and tool life than flood conditions [4,5]. The smaller cutting forces are reported under the MQL condition when compared to dry and flood machining [6,7]. There are many types of cutting fluids, such as distilled water, oil-in-water emulsion, vegetable oil, etc., that have been studied and applied, among which vegetable oils are the most promising alternative solution for the MQL method because they have many advantages, such as high viscosity, biodegradability, non-toxicity and environmentally friendly properties [7,8]. The MQL technique is not only suitable for hard machining but also helps to improve the efficiency of the cutting process [8]. However, the huge amount of heat generated from the cutting zone is still a major challenge, so the application of MQL is still very limited due to the low cooling efficiency, especially for difficult-to-cut materials such as hardened steels, Ni alloy, Ti alloy and so on [9]. Therefore, the selection of proper cutting conditions and cooling lubrication modes plays an important role. In recent years, the minimum quantity cooling lubrication (MQCL) method has been studied and developed in order to overcome the low cooling effect of MQL. The MQCL method also delivers a small amount of lubricant in mist form into the cutting zone, but the cutting fluid used in MQCL has the cooling property to reduce the temperature [10]. MQCL has shown the ability to lubricate and cool more effectively than dry and wet machining in the turning of Ti6Al4V alloy [11]. The mist formation of oil-in-water emulsion droplets in MQCL hard machining was also reported in the hard turning of AISI 1045 steel [12]. The improvement in cooling and lubrication effects contributes to reducing friction and tool wear. The influence of MQCL performance on chip deformation in the cutting zone is analyzed in [13]. The results of the chip morphology and size analysis show the effectiveness of MQCL in cooling lubrication when turning austenitic stainless steel 316L. Moreover, the surface quality and surface layer structure are better under MQCL than those in dry conditions. The reason is that low-temperature oil droplets are formed, providing a superior cooling effect, thereby reducing surface deformation. The diameter and number of oil droplets are strongly influenced by the nozzle distance, air flow rate and air pressure [14], and they can be controlled by the air flow rate and nozzle distance. The study result suggested the reasonable conditions for oil mist formation, and the oil droplets are evaporated when they come into contact with a high-temperature surface in a short time [15]. The optimal cutting parameters were determined for the milling process of Ti-6Al-4V alloy using the MQCL method [16].

However, most of the studies mainly focused on using cutting oils for reducing the friction and temperature in the cutting zone, so it is necessary to investigate the appropriate cutting and cooling lubrication conditions for the MQCL method. Nano cutting fluids have been known for their higher heat transfer and lubricating ability, so they are being studied for machining applications and very promising results have been reported [17,18]. The effectiveness of nanofluid minimum quantity lubrication (NF MQL) application in the cutting process was analyzed and evaluated in [19]. The research results indicate that NF MQL shows better results compared to MQL with pure-based oil in terms of cutting temperature, dimensional accuracy and surface roughness. Shen et al. [20] studied the effects of diamond and $Al_2O_3$ water-based nanofluids on the grinding process of cast iron. Higher grinding efficiency, lower tangential cutting force, better surface structure and lower grinding temperature were reported when using nanofluids. A study on the effects of the flow rate and concentration of $Al_2O_3$ on the grinding process of Ti-6Al-4 V alloy was conducted to prove that the $Al_2O_3$ nano cutting fluid reduced the grinding temperature and friction, so the grinding performance was improved [21]. The investigation of the MQL hard turning process of ADI, a difficult-to-cut material, using $Al_2O_3$ nanoparticles suspended with vegetable oil compared to dry, flood and MQL with pure oil was reported in [22]. The MQL machining performance using 4.0% $Al_2O_3$ vegetable

oil-based nano cutting oil showed the better results due to the improvement in the thermal conductivity and lubricating property of the oil. Hence, a significant reduction in the friction coefficient leads to a decrease in the cutting forces and extension of the tool life [23,24]. The performance of $Al_2O_3$ nano cutting oil in the machining process of Inconel 600 alloy was studied to point out the significant reductions in cutting force, surface roughness, cutting temperature and tool wear [25]. A similar observation was made in a study on the investigation of $Al_2O_3$ soybean oil-based nanofluid in the end milling of SKD 11 tool steel [26]. The improvement in the thermal conductivity and viscosity of the nanofluid compared to the base fluid is the main reason for the enhancement in the cooling and lubricating performance of the MQL technique, leading to reduced tool wear and increased surface quality. M.K. Gupta et al. [27] performed a study on the effects of $Al_2O_3$, $MoS_2$ and graphite nano cutting fluids on the MQL turning of titanium alloy. The cooling and lubricating performance of the MQL method was improved by using nanofluids. The authors pointed out that each type of nanoparticles creates a lubricating mechanism in the cutting zone. G. Gaurav et al. [28] suspended $MoS_2$ nanosheets in jojoba oil, a new type of vegetable oil, and used this for the MQL hard turning of Ti-6Al-4V. The authors concluded that the high viscosity combined with the lamellar structure of $MoS_2$ nanomaterials contributed to improving the machining performance. Moreover, $MoS_2$ nanomaterials exhibit good lubricating properties and a very low friction coefficient. A. H. Elsheikh et al. [29] performed a study on the MQL turning of AISI 4340 alloy using $Al_2O_3$ and CuO nano cutting fluids. The enhancement of the thermophysical properties of rice bran oil was reported by suspending $Al_2O_3$ and CuO nanoparticles, which led to an improvement in the hard turning performance. Moreover, CuO nanofluid showed better results in terms of surface quality and tool wear compared to $Al_2O_3$ nanofluid. The main reasons behind this were the lower contact angle and surface tension of the CuO nanofluid compared with the $Al_2O_3$ nanofluid, which helped to enhance the wettability and spreadability of droplets in the contact faces. Furthermore, 90SiCr is a low-alloy tool steel widely used in mechanical applications such as cold-stamping dies and tools. Due to the content of Si and Cr elements, 90SiCr steel has high hardenability and hardness. Si element can increase the hardness of steel and strengthen the solid solution. At the same time, Si and Cr elements make 90SiCr steel particularly abrasive, so in the hardened state, the hard turning process using carbide inserts under dry conditions faces huge problems. The very high passive force combined with the high temperature in the cutting zone causes early tool wear, leading to a very short tool life [30]. On the other hand, flood coolant has a low lubricating effect and harmful impacts on health and the environment, as well as high costs in treating used cutting fluids [31], so it is necessary to develop eco-friendly cooling and lubricating techniques to address these problems. MQL and MQCL using nano cutting fluids have been proven as promising solutions and gained much attention in recent years. However, studies on the influence of the technological parameters of nanofluid minimum quantity lubrication (NF MQL), nanofluid minimum quantity cooling lubrication (NF MQCL) and cutting conditions on the hard turning outputs are still limited. For these reasons, the authors were motivated to conduct a study on the effects of the cutting fluid type, lubrication method, nanoparticle type, nanoparticle concentration, cutting speed and feed rate on surface roughness in the MQL and MQCL hard turning of 90CrSi (60 ÷ 62 HRC) alloy tool steel using nano cutting oil. The two types of nanoparticles used in this work were $Al_2O_3$ and $MoS_2$. The results of this study will not only provide important technical guidance for using $Al_2O_3$ and $MoS_2$ nano cutting fluids in MQL and MQCL hard turning, but also provide a direction for further studies.

## 2. Materials and Methods

The experimental setup is shown in Figure 1. CS-460 × 1000 Chu Shing lathe (Pin Shin Machinery Co., Ltd., Taichung city, Taiwan) was used for implementing the experiments. The 90CrSi hardened steel samples had a diameter of 40 mm with hardness of 60 ÷ 62 HRC. The chemical composition and mechanical properties of 90 CrSi steel according to the DIN

17350-80 standard are shown in Tables 1 and 2. The hardness of the sample was measured by the Mitutoyo HR-521. CNMG120404 TM T9125 Tungalloy CCD-coated hard alloy inserts (Tungaloy Corporation 11-1 Yoshima-Kogyodanchi Iwaki-city, Fukushima, 970-1144 Japan) were used and their technical specifications are shown in Table 3. Two types of cutting oils used were soybean oil (So) and emulsion oil (Em). $Al_2O_3$ and $MoS_2$ nanoparticles were used to form nano cutting oils. $Al_2O_3$ nanoparticles had a spherical morphology with an average size of 30 nm, manufactured by Soochow Hengqiu Graphene Technology Co., Ltd., Suzhou, China (Figure 2). $MoS_2$ nanoparticles had a layered structure with an average size of 30 nm, manufactured by Luoyang Tongrun Info Technology Co., Ltd., Luoyang, China (Figure 3). Nano cutting oil was prepared by ultrasonic vibration at 40 kHz for 30 ÷ 45 min using the Ultrasons–HD ultrasonicator (JP SELECTA, Abrera, Spain) [32]. After preparation, four different nano cutting oils were obtained: NF So-$Al_2O_3$, NF Em-$MoS_2$, NF Em-$Al_2O_3$ and NF So-$MoS_2$. Noga minicool MC1700 (Noga Engineering & Technology (2008) Ltd., Shlomi, Israel) for the MQL system and a Frigid-X Sub-Zero Vortex tube from Nex Flow™ (Richmond Hill, ON, Canada) for the MQCL system were used. Air pressure P = 6 bar and air flow rate of 200 L/min [33], and cutting depth *t* = 0.15 mm were fixed. The surface roughness values of the machined surface were measured by SJ210 Mitutoyo (Mitutoyo Corporation, Kawasaki, Kanagawa, Japan). After each cutting trial, the surface roughness was measured 3 times and taken as the average value.

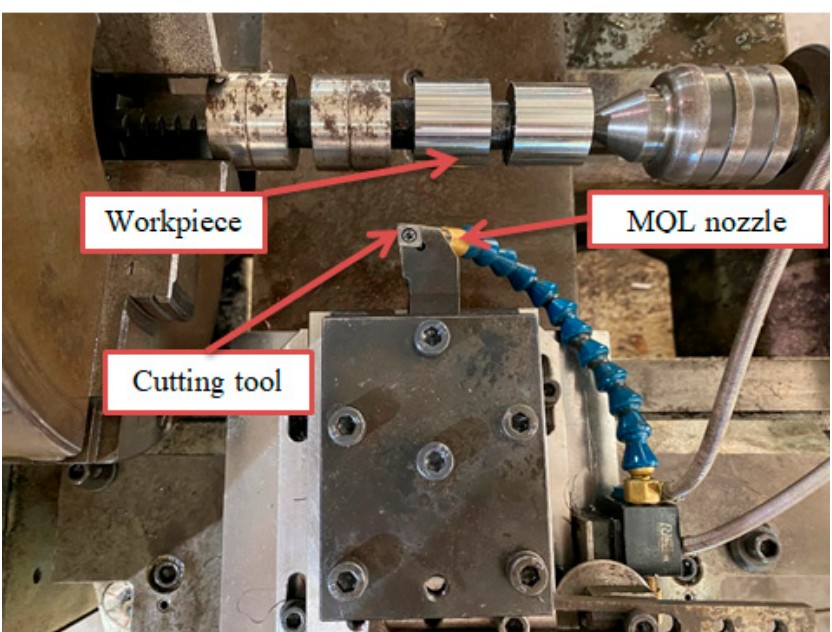

**Figure 1.** Experimental setup.

**Table 1.** Chemical composition in % of 90CrSi steel (DIN 17350-80).

| Element | C | Si | Cr | Cu | Mo | P | Ni | S | Mn | W | V | Ti |
|---|---|---|---|---|---|---|---|---|---|---|---|---|
| **Weight (%)** | 0.85 ÷ 0.95 | 1.20 ÷ 1.60 | 0.95 ÷ 1.25 | Max 0.3 | Max 0.20 | Max 0.03 | Max 0.40 | Max 0.03 | 0.30 ÷ 0.60 | Max 0.20 | Max 0.15 | Max 0.03 |

**Table 2.** Mechanical properties under T = 20 °C of 90 CrSi steel.

| Properties | Tensile Strength $\sigma_B$ (Mpa) | Yield Stress $\sigma_T$ (Mpa) | Specific Elongation at Fracture δ5 (%) | Reduction of Area ψ (%) | Impact Strength KCU (kJ/m²) |
|---|---|---|---|---|---|
| **Values** | 790 | 445 | 26 | 54 | 390 |

**Table 3.** Technical specification of tungalloy CCD-coated hard alloy inserts (CNMG120404 TM T9125).

| Insert Included Angle (°) | Clearance Angle Major (°) | Cutting Edge Length (Mm) | Face Land Width (Mm) | Insert Rake Angle (°) | Corner Radius (Mm) | Chip Breaker Type | Coating Main Composition | Thickness (μm) |
|---|---|---|---|---|---|---|---|---|
| 800 | 0 | 12.9 | 26 | 20 | 0.4 | TM | TiCN-Al2O3 | 16 |

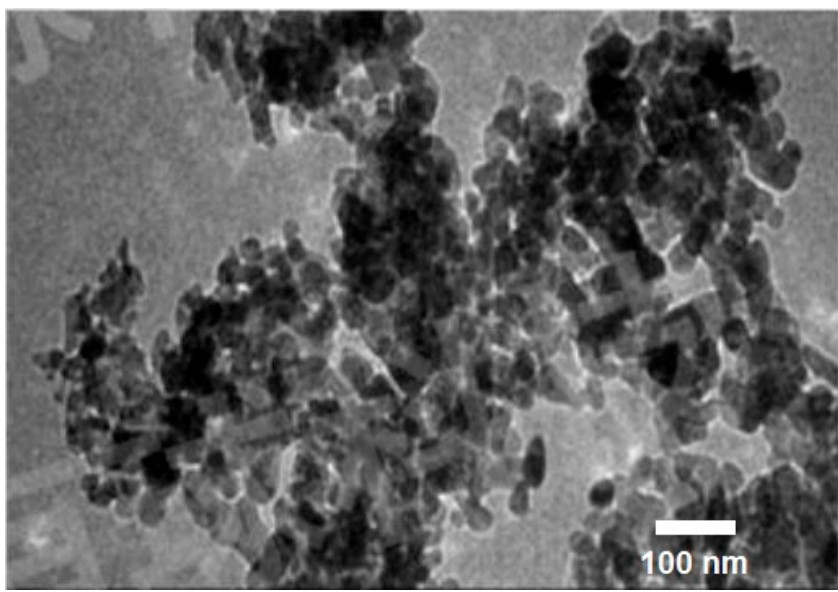

**Figure 2.** Transmission electron microscopy (TEM) image of $Al_2O_3$ nanoparticles. Adapted from [32].

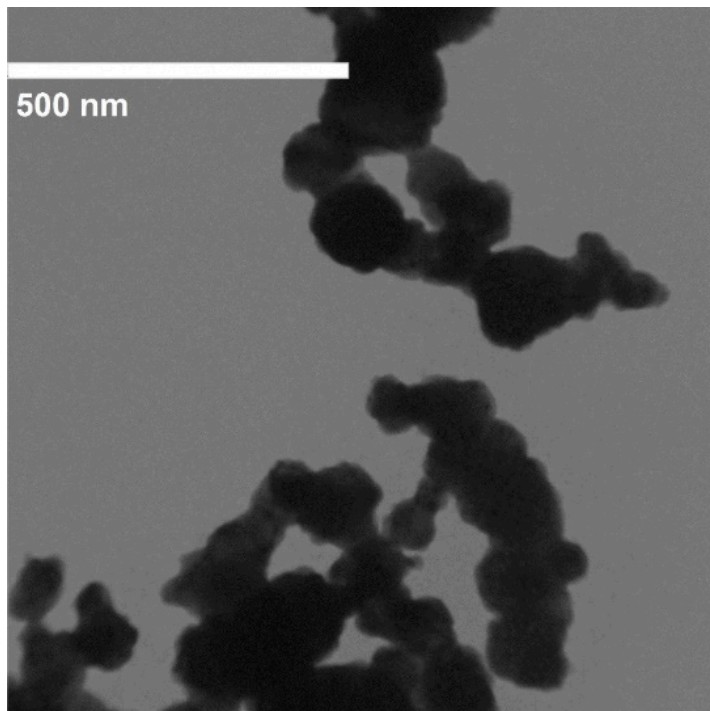

**Figure 3.** TEM image of $MoS_2$ nanosheets. Adapted from [32].

The $2^{k-p}$ experimental design with 6 input variables was used to evaluate the influence of fluid type (FT), lubrication method (LM), nanoparticle type (NP), nanoparticle concentration (NC), cutting speed (V) and feed rate (f) on surface roughness $R_a$ (Table 4) and their value levels were based on the previous study [32].

**Table 4.** Input parameters and their types/levels.

| No. | Input Variables | Symbol | Type/Level | | Response Variable |
|---|---|---|---|---|---|
| | | | Low Level | High Level | |
| 1 | Fluid type (FT) | A | Emulsion (Em) | Soybean oil (So) | |
| 2 | Lubrication methods (LM) | B | MQL | MQCL | |
| 3 | Nanoparticle type (NP) | C | $Al_2O_3$ | $MoS_2$ | |
| 4 | Nanoparticle concentration (NC), wt% | D | 1% | 3% | Surface roughness $R_a$ |
| 5 | Cutting speed (V), m/min | E | 80 | 160 | |
| 6 | Feed rate (f), mm/rev | F | 0.1 | 0.2 | |

Design Expert 11 software was used for the experimental design, and the experimental matrix was built with 32 trials. The experimental trials were performed by following the experimental design, and $R_a$ values were measured after each trial by Mitutoyo SJ210 and the obtained results are shown in Table 5.

**Table 5.** Experimental design.

| Std | Run | FT | LM | NP | NC (wt%) | V (m/min) | F (mm/rev) | $R_a$ (µm) |
|---|---|---|---|---|---|---|---|---|
| 13 | 1 | Em | MQCL | $MoS_2$ | 1 | 80 | 0.1 | 0.578 |
| 12 | 2 | So | MQL | $MoS_2$ | 1 | 80 | 0.2 | 1.485 |
| 8 | 3 | So | MQCL | $Al_2O_3$ | 1 | 80 | 0.2 | 0.899 |
| 23 | 4 | So | MQCL | $Al_2O_3$ | 3 | 80 | 0.1 | 0.822 |
| 28 | 5 | So | MQL | $MoS_2$ | 3 | 80 | 0.1 | 0.625 |
| 21 | 6 | Em | MQCL | $Al_2O_3$ | 3 | 160 | 0.1 | 0.680 |
| 22 | 7 | Em | MQCL | $Al_2O_3$ | 3 | 160 | 0.1 | 0.859 |
| 18 | 8 | Em | MQL | $Al_2O_3$ | 3 | 80 | 0.2 | 1.225 |
| 3 | 9 | So | MQL | $Al_2O_3$ | 1 | 160 | 0.1 | 0.406 |
| 16 | 10 | So | MQCL | $MoS_2$ | 1 | 160 | 0.1 | 0.869 |
| 14 | 11 | Em | MQCL | $MoS_2$ | 1 | 80 | 0.1 | 0.554 |
| 9 | 12 | Em | MQL | $MoS_2$ | 1 | 160 | 0.2 | 1.447 |
| 17 | 13 | Em | MQL | $Al_2O_3$ | 3 | 80 | 0.2 | 1.216 |
| 24 | 14 | So | MQCL | $Al_2O_3$ | 3 | 80 | 0.1 | 0.698 |
| 32 | 15 | So | MQCL | $MoS_2$ | 3 | 160 | 0.2 | 1.466 |
| 27 | 16 | So | MQL | $MoS_2$ | 3 | 80 | 0.1 | 0.570 |
| 20 | 17 | So | MQL | $Al_2O_3$ | 3 | 160 | 0.2 | 1.074 |
| 5 | 18 | Em | MQCL | $Al_2O_3$ | 1 | 160 | 0.2 | 0.654 |
| 31 | 19 | So | MQCL | $MoS_2$ | 3 | 160 | 0.2 | 1.105 |
| 26 | 20 | Em | MQL | $MoS_2$ | 3 | 160 | 0.1 | 0.774 |
| 7 | 21 | So | MQCL | $Al_2O_3$ | 1 | 80 | 0.2 | 0.894 |
| 1 | 22 | Em | MQL | $Al_2O_3$ | 1 | 80 | 0.1 | 0.626 |
| 30 | 23 | Em | MQCL | $MoS_2$ | 3 | 80 | 0.2 | 0.981 |

**Table 5.** *Cont.*

| Std | Run | FT | LM | NP | NC (wt%) | V (m/min) | F (mm/rev) | $R_a$ (μm) |
|-----|-----|-----|------|--------|----------|-----------|------------|-----------|
| 6 | 24 | Em | MQCL | $Al_2O_3$ | 1 | 160 | 0.2 | 1.024 |
| 29 | 25 | Em | MQCL | $MoS_2$ | 3 | 80 | 0.2 | 0.977 |
| 4 | 26 | So | MQL | $Al_2O_3$ | 1 | 160 | 0.1 | 0.405 |
| 2 | 27 | Em | MQL | $Al_2O_3$ | 1 | 80 | 0.1 | 0.633 |
| 15 | 28 | So | MQCL | $MoS_2$ | 1 | 160 | 0.1 | 0.971 |
| 10 | 29 | Em | MQL | $MoS_2$ | 1 | 160 | 0.2 | 1.433 |
| 11 | 30 | So | MQL | $MoS_2$ | 1 | 80 | 0.2 | 1.511 |
| 25 | 31 | Em | MQL | $MoS_2$ | 3 | 160 | 0.1 | 0.755 |
| 19 | 32 | So | MQL | $Al_2O_3$ | 3 | 160 | 0.2 | 0.850 |

## 3. Results

ANOVA analysis with the significance level $\alpha = 0.05$ was carried out by using Design Expert 11, and the results are shown in Table 6. The influence levels of the investigated variables and their interactions on response values $R_a$ are given in Figure 4. The suitability of the survey model and the set of experimental parameters are shown in Figure 5. The independent influence of the input variables on Ra is shown in Figure 6. Interaction effects on $R_a$ are given in Figure 7.

**Table 6.** Results of ANOVA analysis for surface roughness.

| Source | Sum of | DF | Mean | F-Value | *p*-Value |
|--------|--------|-----|----------|-----------|-----------|
| Model | 2.972518 | 13 | 0.228655 | 20.95669 | 0.0001 |
| FT | 0.001711 | 1 | 0.001711 | 0.156828 | 0.696749 |
| LM | 0.031501 | 1 | 0.031501 | 2.887081 | 0.106509 |
| NP | 0.307328 | 1 | 0.307328 | 28.1672 | 0.0001 |
| NC | 0.002592 | 1 | 0.002592 | 0.237562 | 0.631856 |
| V | 0.00714 | 1 | 0.00714 | 0.654406 | 0.429105 |
| f | 1.718658 | 1 | 1.718658 | 157.5183 | 0.0001 |
| FT*LM | 0.21125 | 1 | 0.21125 | 19.36147 | 0.000345 |
| FT*NP | 0.121525 | 1 | 0.121525 | 11.13795 | 0.003666 |
| FT*NC | 0.017485 | 1 | 0.017485 | 1.602488 | 0.221691 |
| FT*V | 0.044551 | 1 | 0.044551 | 4.083196 | 0.058444 |
| FT*f | 0.005513 | 1 | 0.005513 | 0.505231 | 0.48632 |
| LM*NP | 0 | 0 | - | - | - |
| LM*NC | 0.12525 | 1 | 0.12525 | 11.47941 | 0.003277 |
| LM*V | 0 | 0 | - | - | - |
| LM*f | 0.378015 | 1 | 0.378015 | 34.64581 | 0.0001 |
| NP*NC | 0 | 0 | - | - | - |
| NP*V | 0 | 0 | - | - | - |
| NP*f | 0 | 0 | - | - | - |
| NC*V | 0 | 0 | - | - | - |
| NC*f | 0 | 0 | - | - | - |
| V*f | 0 | 0 | - | - | - |

**Table 6.** *Cont.*

| Source | Sum of | DF | Mean | F-Value | *p*-Value |
|---|---|---|---|---|---|
| Residual | 0.196395 | 18 | 0.010911 | - | - |
| Lack of Fit | 0.006283 | 2 | 0.003142 | 0.264402 | 0.770954 |
| Pure Error | 0.190112 | 16 | 0.011882 | - | - |
| Cor Total | 3.168913 | 31 | - | - | - |

"*" represents the interactions between the factors.

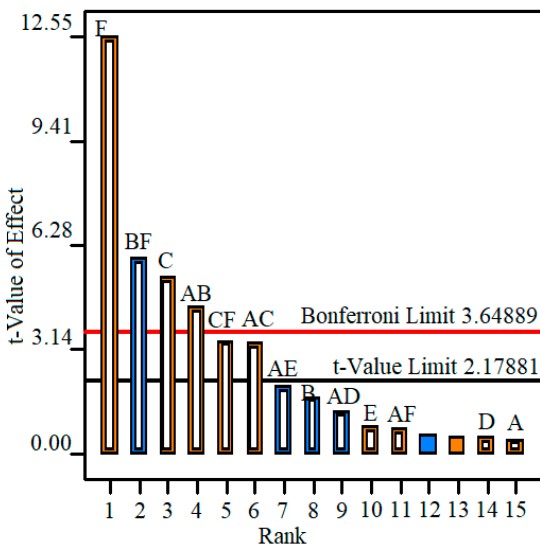

**Figure 4.** Pareto chart of the effects of the input variables on surface roughness (A—fluid type; B—lubrication methods; C—nanoparticle type; D—nanoparticle concentration; E—cutting speed; F—feed rate).

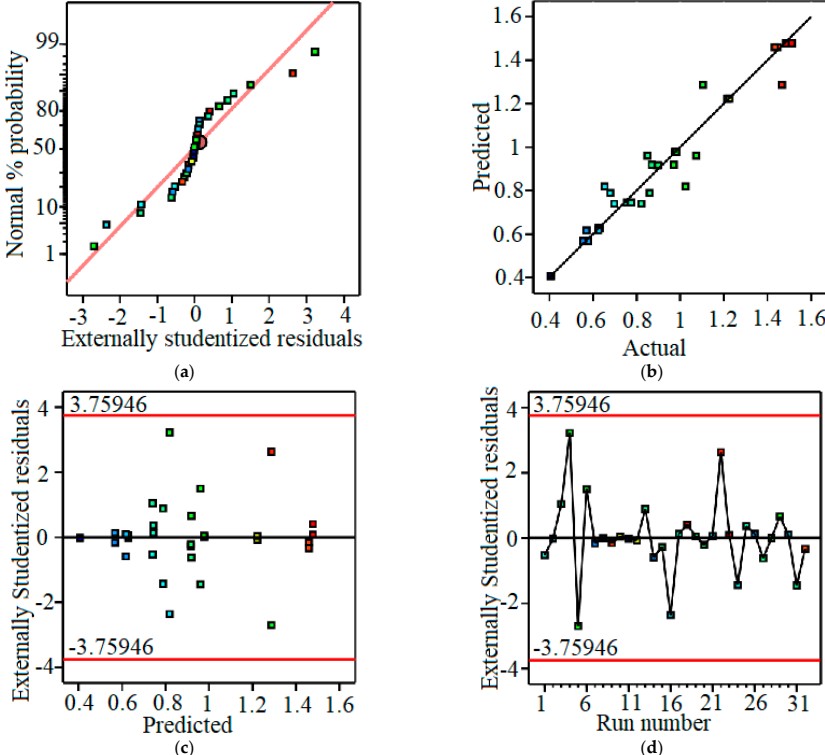

**Figure 5.** Model fit chart: (**a**)—normal plot of residuals (the reference line is red); (**b**)—predicted vs. actual (the reference line is black); (**c**)—residual vs. predicted (the limit line is red and the reference line is black); (**d**)—residual vs. run (the limit line is red and the reference line is black).

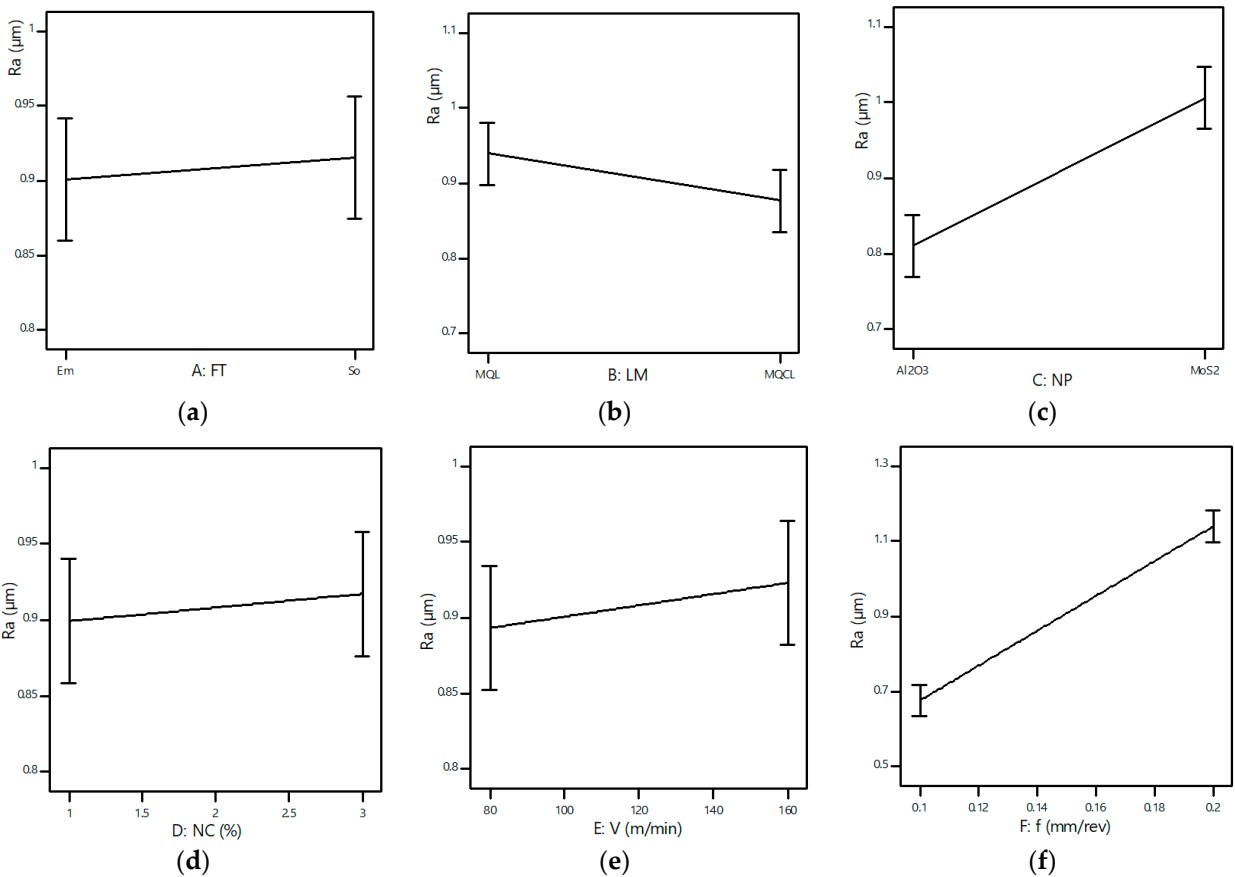

**Figure 6.** The main effects of input parameters: (**a**)—FT; (**b**)—LM; (**c**)—NP; (**d**)—NC (%); (**e**)—V (m/min); (**f**)—f (mm/rev).

In this study, a first-order model (2FI) is used to analyze the influence of input variables and their interactions on surface roughness. The ANOVA results indicate that in the studied range, the feed rate (f), the type of nanoparticles (NP) and the interactions FT*LM, FT*NP, LM*NC and LM*f have *p*-values less than 0.05 and large Fisher coefficients, proving that these survey factors are significant. Other factors and interactions among them have large *p*-values, so they have little significance for $R_a$ (Table 6).

The Pareto chart in Figure 4 shows the influence of the input variables and their interaction on surface roughness. Studying the Bonferroni and t-Value limit lines, the investigated factors and their interactions over the t-Value limit line have effects on $R_a$ and have a significant influence if they exceed the Bonferroni limit line. Thus, F, B*F, C, A*B have strong effects on surface roughness, in which the feed rate f causes the strongest influence, followed by the type of nanoparticles. The interactions of the lubrication method and the feed rate (B*F) as well as the fluid type and the lubrication method (A*B) also significantly affect the surface roughness. Lubrication methods (B), cutting speed (E), fluid type, nanoparticle concentration (D) and the interactions A*D, A*E have little influence on $R_a$, and the other ones have almost no effects.

The results of the evaluation of the fit of the used model are shown in Figure 5. Figure 5a depicts the normal plot, which shows that the residuals of the experimental points are all located along the reference line, so the residuals follow a normal distribution. Figure 5b shows that the predicted values of surface roughness are quite close to the experimental roughness values when using the predicted regression model. The residual versus prediction chart (Figure 5c) and residual plot at experimental points (Figure 5d) show that the calculated values are in the limit region; in other words, the selected model is suitable and there is no need to perform model conversion. Thus, using a first-order

model with interaction between two factors to evaluate the influence of input parameters on surface roughness value is appropriate and statistically significant.

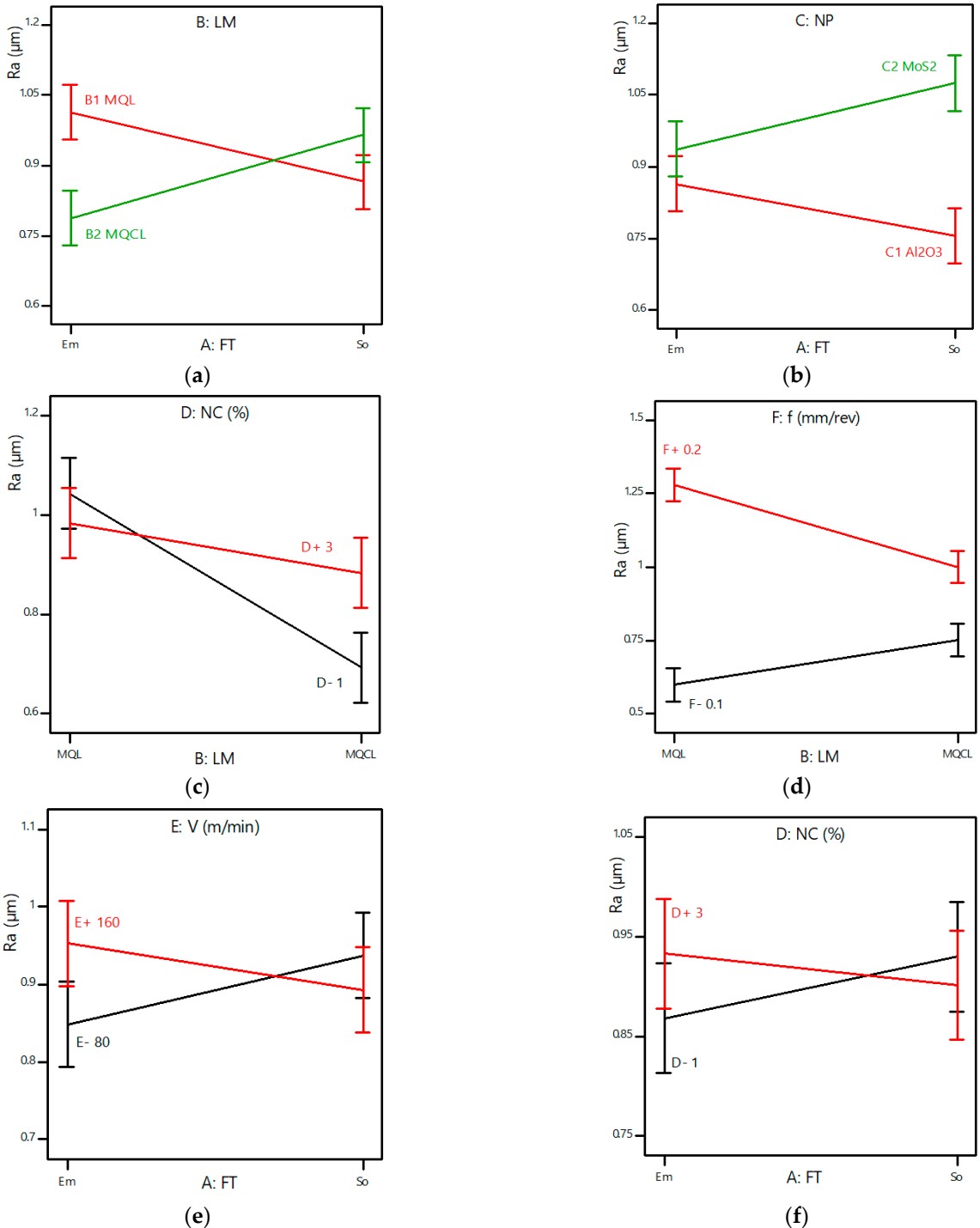

**Figure 7.** Interaction effects between input parameters on surface roughness: (**a**)—FT*LM; (**b**)—FT*NP; (**c**)—LM*NC; (**d**)—LM*f; (**e**)—FT*V; (**f**)—FT*NC.

The influence of each investigated variable on surface roughness in the hard turning of 90CrSi steel with different technological conditions is shown in Figure 6. In Figure 6a, the average values of surface roughness with soybean oil are larger than those with emulsion oil. The reason is that soybean oil has a lower ignition temperature (approximately 450 °F (232.2 °C)), so it is burned when the cutting temperature is high and loses its lubricating ability [33]. However, adding nanoparticles to soybean oil contributes to increasing

the viscosity and thermal conductivity of the base oil, thereby improving the cooling and lubricating characteristics [34]. As shown in Figure 6b, the application of MQCL gives smaller values of surface roughness than those under MQL due to the superior cooling performance.

As shown in Figure 6c, the type of nanoparticles has a great influence on surface roughness. $R_a$ values with $Al_2O_3$ nanofluid (NF) are smaller than those with $MoS_2$ NF because $Al_2O_3$ nanoparticles not only have a good lubricating property, but also possess good thermal conductivity, which enhances the heat resistance of the nano cutting oil and improves the lubricating efficiency [35]. In addition, $Al_2O_3$ nanoparticles have higher hardness and a spherical morphology, which create a "ball roller" mechanism in the cutting zone, reducing the friction coefficient and the tool scratches, so the surface roughness is reduced [36]. Meanwhile, $MoS_2$ nanosheets have a layered structure, so they only create a tribo-film mechanism, and their main effect is to reduce friction.

The nanoparticle concentration also affects the surface roughness, and the concentration of 1.0 wt% gives better results than 3.0 wt% (Figure 6d) because 1.0 wt% is the appropriate concentration of $Al_2O_3$ nanoparticles [37]. When the $Al_2O_3$ nanoparticle concentration rises 3.0 wt%, the impedance and collision phenomenon will occur and produce inconsistent lubrication. Therefore, $Al_2O_3$ nanoparticles with high hardness, acting as "tiny cutters", will scratch and deteriorate the machined surface, thus increasing the $R_a$ values [35]. To increase the concentration of $MoS_2$, more $MoS_2$ nanoparticles adhere to the cutting edge and cause surface scratches, so the $R_a$ values are increased [3,38]. However, further studies are needed to investigate and determine the optimal nanoparticle concentration to achieve the minimal $R_a$ value.

As shown in Figure 6e, the average surface roughness value at a speed of 80 m/min is smaller than that at a high speed of 160 m/min. The reason is that in hard machining, the cutting forces rise with the increasing cutting speed [36], causing vibration, which increases the surface roughness value [15]. However, under MQL or MQCL conditions using nano cutting fluids, one can use a greater cutting speed compared to the conventional hard turning process, while maintaining the surface roughness value. In order to clearly analyze the effect of cutting speed under different conditions and determine the optimal value, more research is needed.

As shown in Figure 6f, the $R_a$ values rise with the increase in feed rate, which proves the strongest influence of geometric factors in the hard turning process [39], but in order to determine the interaction effect between the feed rate and other factors and to evaluate the machining efficiency under roughing or finishing conditions, it is necessary to analyze the effect of the feed rate on the surface roughness values.

The interaction effect between fluid type and lubrication method (FT * LM): it can be observed that $R_a$ is greatly affected by the lubrication method and fluid type. Emulsion oil yields better results than soybean oil under the MQL condition, but for MQCL, using emulsion oil yields smaller surface roughness values (Figure 7a) because it has low viscosity, meaning that it can easily split into small droplets to penetrate the cutting zone. In addition, the viscosity of emulsion oil increases with the low-temperature effects of MQCL [3].

The interaction effect between fluid type and nanoparticle type (FT*NP): as shown in Figure 7b, the type of nanoparticles suspended in the based cutting oil changes the cooling and lubricating mechanism in the cutting zone. When using the $MoS_2$ emulsion oil-based nanofluid, the surface roughness is smaller than that with the $MoS_2$ soybean oil-based nanofluid. For $Al_2O_3$ nanoparticles suspended in soybean oil, $R_a$ is smaller. The main reason is that $MoS_2$ nanoparticles have a layered structure, so they easily form the tribo-film, contributing to the lubricating performance of the emulsion oil. Soybean oil has higher viscosity but a low ignition temperature, and when mixed with $Al_2O_3$ nanoparticles, its thermal conductivity improves. Combined with the good lubricating property of $Al_2O_3$ nanoparticles, this contributes to reducing the surface roughness [26]. Moreover, soybean oil has a chain fatty acidic structure, which contributes to its better lubricating characteristic than emulsion oil [29].

The interaction effect between lubrication method and nanoparticle concentration (LM*NC): as shown in Figure 7c, when the nanoparticle concentration rises from 1.0 wt% to 3.0 wt%, the surface roughness values change very little under the MQL condition and increase dramatically under MQCL. MQCL shows better results than MQL.

The interaction effect between lubrication method and feed rate (LM*f): in Figure 7d, using a small feed rate under MQL gives better surface roughness than MQCL. For a higher feed rate of 0.2 mm/rev, MQCL presents the better results due to the superior cooling effect, which is effective in reducing the growing cutting temperature caused by the increasing feed rate.

The interaction effect between fluid type and cutting speed (FT*V): according to Figure 7e, when the cutting speed is increased from 80 m/min to 160 m/min, the surface roughness changes differently with different types of cutting oil. For emulsion oil, the surface roughness increases significantly with increasing cutting speed. In contrast, when using soybean oil, the surface roughness decreases slightly with increasing cutting speed.

The interaction effect between fluid type and nanoparticle concentration (FT*NC): as shown in Figure 7f, the interaction between fluid type and nanoparticle concentration has little effect on surface roughness. A lower nanoparticle concentration should be used in emulsion oil and shows better performance compared to soybean oil. The average surface roughness values did not change significantly with increasing nanoparticle concentration.

Through the study of the interaction effects above, it is possible to highlight the research direction in the selection of input parameters. To reduce the surface roughness, one requires the combination of MQCL with emulsion oil, $MoS_2$ nanoparticles, a low nanoparticle concentration of 1.0 wt%, low feed rate (0.1 mm/rev) and low cutting speed (80 m/min). Meanwhile, one should use the MQL mode combined with soybean oil, $Al_2O_3$ nanoparticles, a low nanoparticle concentration (1.0 wt%) and low levels of cutting speed and feed rate at 80 m/min and 0.1 mm/rev, respectively, to achieve lower $R_a$ values. However, the cutting temperature was not investigated, in order to clearly show the cooling and lubricating effects of lubrication modes using different nano cutting oils. Moreover, the cooling and lubricating properties of soybean oil were improved by suspending $Al_2O_3$ nanoparticles, so its applicability will be enlarged for hard turning; thus, it represents an alternative solution for some of the grinding processes and retains the environmentally friendly characteristics of the MQL and MQCL methods.

## 4. Conclusions

In this study, the influences of fluid type, lubrication method, nanoparticle types, nanoparticle concentration, cutting speed and feed rate on surface roughness in the MQL and MQCL hard turning of 90CrSi alloy tool steel are investigated by using ANOVA analysis applied for a $2^{k-p}$ experimental design. To enhance the cooling and lubricating performance for the hard turning process, the MQL and MQCL techniques using $Al_2O_3$ and $MoS_2$ nano cutting fluids were utilized. The main contributions of this study can be summarized as follows.

The machinability of carbide inserts, which are recommended for steels before heat treatment, is much improved and enlarged for hard turning by using new, environmentally friendly cooling and lubricating methods, namely MQL and MQCL, using nanofluids. The obtained results indicate that this type of carbide insert can be effectively used for hard machining, and the highest machinable hardness rises from 35 HRC to 60 ÷ 62 HRC with the same cutting conditions recommended by manufacturers. This will result in alternative solutions and economic benefits.

The MQCL method shows a higher cooling effect, which enhances the heat dissipation in the cutting zone compared to the MQL technique. Hence, the surface quality is improved.

$Al_2O_3$ nanofluid contributes to achieving lower surface roughness values compared to $MoS_2$ nanofluid. However, using the appropriate nanoparticle concentration for each nano cutting oil is a very important factor and has a great influence on the results of the

machining process. Therefore, further studies are needed to investigate and optimize this parameter.

Feed rate exerts the strongest influence on surface roughness in hard machining, followed by the type of nanoparticles, while fluid type, nanoparticle concentration and cutting speed show the lowest impacts. The interactions of fluid type and lubrication method (FT*LM); fluid type and nanoparticle type (FT*NP); lubrication method and nanoparticle concentration (LM*NC); and lubrication method and feed rate (LM*f) have significant impacts on $R_a$, which indicates the research direction for further studies.

The analysis results show that the MQCL method using $MoS_2$ nanoparticles of 1.0 wt% suspended in emulsion oil with V = 80 m/min, f = 0.1 mm/rev or the MQL method using $Al_2O_3$ nanoparticles of 1.0 wt% suspended in soybean oil with V = 80 m/min, f = 0.1 mm/rev can effectively reduce the surface roughness values in the hard turning of 90 CrSi steel.

In further studies, more investigations should be carried out to determine the optimized values and levels for each lubrication method.

**Author Contributions:** Conceptualization, T.M.D., N.M.T. and T.T.L.; methodology, V.L.H.; software, V.L.H. and T.B.N.; data curation, T.B.N.; writing—original draft preparation, T.M.D.; writing—review and editing, T.T.L. and T.B.N.; visualization, N.M.T.; supervision, T.M.D. All authors have read and agreed to the published version of the manuscript.

**Funding:** The work presented in this paper is funded by Thai Nguyen University of Technology, Thai Nguyen University, Vietnam.

**Acknowledgments:** The work presented in this paper is supported by Thai Nguyen University of Technology, Thai Nguyen University, Vietnam. All the authors have consented to the acknowledgement.

**Conflicts of Interest:** The authors declare no conflict of interest.

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
