# Peer review of "Investigation of Machining Performance of MQL and MQCL Hard Turning Using Nano Cutting Fluids"

_fluids, doi:10.3390/fluids7050143_

Round 1
Reviewer 1 Report
This paper studied the machining performance of MQL and MQCL hard turning using Nano-cutting Fluids. The methods used in this paper are reasonable while the results seem valid. Hence, it has merits for publication after some Major Revisions. Some suggestions are presented as follows:
(1) How do the authors define the interaction of variables? It seems that it is defined by the product of two variables. What is the basis for this definition?
(2) In Figs. 4 and 5, the range of changes in Ra is very small, sometimes even less than 0.1μm. Whether such a small optimization is necessary in engineering? Are the results of this paper applied to the workpiece that requires very high machining accuracy?
(3) The quality of the figures in this paper needs to be improved.
Fig. 2: The definitions of symbols A-F are recommended to be noted in the blank area.
Fig. 3: What does the red line and the black line mean? How to distinguish predicted value from actual value? There is too little explanatory information in the figure and legends should be added.
(4) Please improve the language and logic of the article, while avoid occasional grammatical errors. For example: ‘In Figure 4, a, the average values of surface roughness when using soybean oil are larger than using emulsion oil.’
Author Response
RESPONSES TO THE REVIEWER 1
We are very grateful for the reviews provided by the editors and each of the external reviewers of this manuscript. Please see below, our detailed response to comments.
This paper studied the machining performance of MQL and MQCL hard turning using Nano-cutting Fluids. The methods used in this paper are reasonable while the results seem valid. Hence, it has merits for publication after some Major Revisions. Some suggestions are presented as follows:
(1) How do the authors define the interaction of variables? It seems that it is defined by the product of two variables. What is the basis for this definition?
Ans:
Thank you very much. The interaction of variables is the interaction of at least two independent variables on a third dependent variable. The interaction of variables can be considered based on the results of analysis of variance (ANOVA). Interactions between variables can be first-order, second-order or third-order interactions. Within the scope of the study, using a 2k experimental matrix should only consider first-order interactions between two factors.
(2) In Figs. 4 and 5, the range of changes in Ra is very small, sometimes even less than 0.1μm. Whether such a small optimization is necessary in engineering? Are the results of this paper applied to the workpiece that requires very high machining accuracy?
Ans:
Thank you very much. Figures 4 and 5 describe the main effects and the interaction effects between the investigated factors on the surface roughness. The purpose of analyzing main effect trends and interaction influences is to limit the research area and research subjects for further studies. Therefore, some survey variables have little influence on the surface roughness value, which will be removed in the following studies.
Although the change of Ra is very small, in finishing, only a very small improvement of the surface roughness value Ra also has the great practical significance, sometimes changing a level of surface smoothness.The results of this study will be applied to the finishing processing of the mechanical parts after heat treatment, which requires very high accuracy.
(3) The quality of the figures in this paper needs to be improved.
Ans:
Thank you very much. The quality of the figures was improved in the revised manuscript, but the figures were exported from Design Expert software, so the improvement was not much. I hope the reviewer understand this issue
Fig. 2: The definitions of symbols A-F are recommended to be noted in the blank area.
Ans:
Thank you very much. The definitions of symbols A-F are noted in the corresponding figure to make readers easier to follow.
Fig. 3: What does the red line and the black line mean? How to distinguish predicted value from actual value? There is too little explanatory information in the figure and legends should be added.
Ans:
Thank you very much. For Fig 3a, the reference line is red and the reference line is black in Fig 3b. The red line is the limit line and the black line is the reference line in Figures 3c, d. The actual value points are distributed along the black reference line (Figure 3b), which contains the predicted points. The closer the actual point distribution is to the standard curve, the better the fit of the experimental data set with the predicted model. The explanatory information in the figure and legends was added in the revised manuscript.
(4) Please improve the language and logic of the article, while avoid occasional grammatical errors. For example: ‘In Figure 4, a, the average values of surface roughness when using soybean oil are larger than using emulsion oil.’
Ans:
Thank you very much. This statement was modified by following the reviewer’s comment and marked in red color.

Reviewer 2 Report
The authors present an experimental investigation on hard turning efficiency of 90CrSi (60 ÷ 62HRC) steel using MQL and MQCL conditions using Al2O3 and MoS2 nanocutting fluids. Analysis of variance (ANOVA) was used to study the influence of MQL and MQCL parameters on surface roughness. The obtained results showed that the uses of MoS2 oil-in-water emulsion-based nanofluid under MQCL or mixed with Al2O3 soybean oil-based nanofluid under MQL can significantly reduce the surface roughness values. The interaction effects between the control variables were also studied to provide technical guidance for further studies using Al2O3 and MoS2 nano cutting fluids as well as the application in machining practice
The paper is highly suitable for publication in FLUIDS, a major revision is recommended for improving the manuscript. I will be honored to review a revised version of the present paper.
- The paper should be restructured and written in a clear manner using a high scientific style.
- English should be improved and checked by a native English speaker.
- Literature review should be improved.
- Missing research gap and the objectives of the paper should be clearly presented.
- XRD, SEM and TEM of the used nanoparticles or nanofluid should be included.
- “Soybean oil has a higher viscosity but low ignition temperature, and when mixed with Al2O3 nanoparticles, its thermal conductivity improves combined with good lubricating property of Al2O3 nanoparticles, which reduce the surface roughness”, more discussion should be given to support this claim.
- Thermophysical properties of all used nanofluids as well as the base fluid should be included.
- What is the effect of those cutting conditions on tool wear?
- Literature should be supported by more publications from FLUIDS, METALS and MATERIALS.
- Use the following performance assessment and chip morphology evaluation of austenitic stainless steel under sustainable machining conditions, temperature field sensing of a thin-wall component during machining: numerical and experimental investigations, a new optimized predictive model based on political optimizer for eco-friendly mql-turning of aisi 4340 alloy with nano-lubricants and others.
- What is coefficient of correlation for all qq plots? (Figure 3)
- The nanoparticle concentration also affects the surface roughness, and the concentration of 1.0 wt% gives better results than 3.0 wt% . It can be explained that the high concentration of Al2O3 nanoparticles causes the impedance and collision phenomenon, and also scratch on the machined surface, thus increasing Ra values.
Can you support your claim with EDX?
Author Response
RESPONSES TO THE REVIEWER 2
We are very grateful for the reviews provided by the editors and each of the external reviewers of this manuscript. Please see below, our detailed response to comments.
The authors present an experimental investigation on hard turning efficiency of 90CrSi (60 ÷ 62HRC) steel using MQL and MQCL conditions using Al2O3 and MoS2 nanocutting fluids. Analysis of variance (ANOVA) was used to study the influence of MQL and MQCL parameters on surface roughness. The obtained results showed that the uses of MoS2 oil-in-water emulsion-based nanofluid under MQCL or mixed with Al2O3 soybean oil-based nanofluid under MQL can significantly reduce the surface roughness values. The interaction effects between the control variables were also studied to provide technical guidance for further studies using Al2O3 and MoS2 nano cutting fluids as well as the application in machining practice
The paper is highly suitable for publication in FLUIDS, a major revision is recommended for improving the manuscript. I will be honored to review a revised version of the present paper.
- The paper should be restructured and written in a clear manner using a high scientific style.
Ans:
Thank you very much. The paper was revised carefully by following the reviewer’s comments. The changes were marked in red color.
- English should be improved and checked by a native English speaker.
Ans:
Thank you very much. The English was revised carefully by following the reviewer’s comment. The changes were marked in red color.
- Literature review should be improved.
Ans:
Thank you very much. The literature review was expanded and the high-quality revelant papers were added to formulate the problem. The changes were marked in red color.
- Missing research gap and the objectives of the paper should be clearly presented.
Ans:
Thank you very much for your valueable comments. The research gap and the objectives of the paper were revised and modified in the revised manuscript.
- XRD, SEM and TEM of the used nanoparticles or nanofluid should be included.
Ans:
Thank you very much. TEM images of the used nanoparticles were added in the revised manuscript.
- “Soybean oil has a higher viscosity but low ignition temperature, and when mixed with Al2O3 nanoparticles, its thermal conductivity improves combined with good lubricating property of Al2O3 nanoparticles, which reduce the surface roughness”, more discussion should be given to support this claim.
Ans:
Thank you very much for the valuable comments. The viscosity and thermal conductivity of soybean oil with/without Al2O3 nanoparticles were measured in our previous study (please see Ref. 26), and the ignition temperature of soybean oil about 450°F (232.2 °C)) (please see Ref. 33). These claims and relevant references were added in the revised manuscript.
- Thermophysical properties of all used nanofluids as well as the base fluid should be included.
Ans:
Thank you very much for the valuable comments. In the content of this article, the author focuses on studying the influence of the input parameters to have the further research direction. The thermophysical property is the interesting topics and has been studied and investigated by many studies. Therefore, the authors ask the reviewer's permission to cite in the revised manuscript.
- What is the effect of those cutting conditions on tool wear?
Ans:
Thank you very much for the valuable comments. In the content of this article, the author focuses on studying the influence of the input parameters on surface roughness. The study on cutting forces, tool wear and tool life will be discussed in the next study.
- Literature should be supported by more publications from FLUIDS, METALS and MATERIALS.
Ans:
Thank you very much. The literature and discussion were cited in the more publications from FLUIDS, METALS, and LUBRICANTS, which were added in the revised manuscript.
- Use the following performance assessment and chip morphology evaluation of austenitic stainless steel under sustainable machining conditions, temperature field sensing of a thin-wall component during machining: numerical and experimental investigations, a new optimized predictive model based on political optimizer for eco-friendly mql-turning of aisi 4340 alloy with nano-lubricants and others.
Ans:
Thank you very much. The word was revised and cited in the revised manuscript.
- What is coefficient of correlation for all qq plots? (Figure 3)
Ans:
Thank you very much. The coefficient of correlation all qq plots? (Figure 3) was automatically chosen by >0.7 from Design Expert 11 software.
- The nanoparticle concentration also affects the surface roughness, and the concentration of 1.0 wt% gives better results than 3.0 wt% . It can be explained that the high concentration of Al2O3 nanoparticles causes the impedance and collision phenomenon, and also scratch on the machined surface, thus increasing Ra values.
Can you support your claim with EDX?
Ans:
Thank you very much for your valuable comments. The authors ask the reviewer's permission to cite the relevant papers. For this topic, the authors are studying the effects of nanoconcentration on the machined surface, but the work is only for hard milling process (please see the figure below). The work will continue to hard turning and discuss in the next paper. The EDX of machined surface is the important and interesting topics, which will be done in the next work after having the optimized values of input parameters. Thank you again.
|
(a) 3D view |
(b) 2D view |
|
Figure 16. Machined surface microstructure under MQL hard milling using 1.5% Al2O3 nanoparticles with p = 6 bar, Q=200 l/min: (a) 3D view, (b) 2D view |
|

Reviewer 3 Report
Thank you for submitting your paper. The work done here draws attention to a significant subject of using MQL in machining. My comments including major and minor concerns are given below:
- Please consider reviewing the abstract and highlight the novelty, major findings, and conclusions. I suggest reorganizing the abstract, highlighting the novelties introduced. The abstract should contain answers to the following questions:
- What problem was studied and why is it important?
- What methods were used?
- What conclusions can be drawn from the results? (Please provide specific results and not generic ones).
- The abstract must be improved. It should be expanded. Please use numbers or % terms to clearly shows us the results in your experimental work.
- Please consider reporting on studies related to your work from mdpi journals.
- The abstract does not read well at all, first any abbreviations must be clearly explained first time they appear in the text. Next, the authors confusing the authors with several terminologies. What is MQCL? The abstract is mainly generic, first several lines can be shortened or completely removed without affecting the abstract content.
- Line 25 needs checking for grammar and spell check.
- Line 27 reduce by how much?
- Line 27 what interaction effects? Did the authors use design of experiments and statistical analysis?
- The introduction must be expanded, please consider improving the introduction, provide more in-depth critical review about past studies similar to your work, mention what they did and what were their main findings then highlight how does your current study brings new difference to the field.
- Add details of the nano particle types used in the study. What is the difference between them.
- Please combine all small paragraphs into larger ones.
- Line 122 check grammar, the paper requires extensive English editing.
- Materials and method section lacks any details of the experimental setup, measurement process used for evaluating surface roughness (images, graphs and measurement setup).
- Table 2 belongs to Materials and method section and not in results section, please move to above.
- Lines 146-148 move to materials and methods section.
- All of a sudden, the authors mention ANOVA in line 150 although it was not mentioned in the abstract.
- Lines 156-157 repetitive, please rephrase.
- I see no scientific discussion at all, just describing the statistical results of the tests.
- Lines 205-208 so what is the thermal conductivity of the two nano particles, mention it to support your claims.
- Lines 208-210 same as above.
- Lines 213-216 contradicting explanation with previous paragraph, before you said hardness of particles is good to reduce roughness, then you said those particles if too many they scratch the machined surfaces and increase roughness, so where is the break point? These are two contradicting results.
- Lines 221-228 does not read well at all, please rephrase and improve the justification. Also, did the authors measure cutting forces to confirm this claim?
- Lines 229-231, the authors again make false claim, they say that Ra increase sharply, but looking at the graphs in figure 4, Ra change is less than 0.2 microns! This is very low and should not be considered signficant at all especially when machining steels.
- Line 244 “brings out the better results than emulsion oil under MQL condition” this sentence does not read well at all.
- The results are merely described and is limited to comparing the experimental observation and describing results. The authors are encouraged to include a more detailed results and discussion section and critically discuss the observations from this investigation with existing literature.
- Conclusion can be expanded or perhaps consider using bullet points (1-2 bullet points) from each of the subsections.
- After reading the paper i found the authors did not make any attempt to provide any comparison of their results with past studies similar or closely related to this work, there is no novelty or sceintific discussion in the discussion. therefore, paper can not be accepted at its current form and authors must extensively improve their article.
Author Response
RESPONSES TO THE REVIEWER 3
We are very grateful for the reviews provided by the editors and each of the external reviewers of this manuscript. Please see below, our detailed response to comments.
Thank you for submitting your paper. The work done here draws attention to a significant subject of using MQL in machining. My comments including major and minor concerns are given below:
- Please consider reviewing the abstract and highlight the novelty, major findings, and conclusions. I suggest reorganizing the abstract, highlighting the novelties introduced. The abstract should contain answers to the following questions:
What problem was studied and why is it important?
What methods were used?
What conclusions can be drawn from the results? (Please provide specific results and not generic ones).
The abstract must be improved. It should be expanded. Please use numbers or % terms to clearly shows us the results in your experimental work.
Ans:
Thank you very much. The abstract was rewriten by following the reviewer’s comments. The changes were marked in red color.
- Please consider reporting on studies related to your work from mdpi journals.
Ans:
Thank you very much. Our work from mdpi journals was added and cited to support the claims and explanation.
- The abstract does not read well at all, first any abbreviations must be clearly explained first time they appear in the text. Next, the authors confusing the authors with several terminologies. What is MQCL? The abstract is mainly generic, first several lines can be shortened or completely removed without affecting the abstract content.
Ans:
Thank you very much. The abstract was rewriten by following the reviewer’s comments. The changes were marked in red color.
- Line 25 needs checking for grammar and spell check.
- Line 27 reduce by how much?
- Line 27 what interaction effects? Did the authors use design of experiments and statistical analysis?
Ans:
Thank you very much. The abstract was rewriten by following the reviewer’s comments. The changes were marked in red color.
- The introduction must be expanded, please consider improving the introduction, provide more in-depth critical review about past studies similar to your work, mention what they did and what were their main findings then highlight how does your current study brings new difference to the field.
Ans:
Thank you very much. The literature review was expanded and the high-quality revelant papers were added to formulate the problem. The changes were marked in red color.
- Add details of the nano particle types used in the study. What is the difference between them.
Ans:
Thank you very much. TEM images and details of the used nanoparticles were added in the revised manuscript.
- Please combine all small paragraphs into larger ones.
Ans:
Thank you very much. The revision was made to change by following the reviewer’s comment.
- Line 122 check grammar, the paper requires extensive English editing.
Ans:
Thank you very much. Line 122 was rewriten and the English was checked carefully and edited in the revised manuscript.
- Materials and method section lacks any details of the experimental setup, measurement process used for evaluating surface roughness (images, graphs and measurement setup).
Ans:
Thank you very much. The surface roughness Ra was measured by 3 times and taken by the average value after each trial by Mitutoyo SJ210, which was added in the revised manuscripts. The image of measurement process was shown as below, but I did not add in the paper.
After each experimental trial, we measured the surface roughness values 3 times and took the average value.
- Table 2 belongs to Materials and method section and not in results section, please move to above.
Ans:
Thank you very much. The table 2 was moved to Materials and method section.
- Lines 146-148 move to materials and methods section.
Ans:
Thank you very much. Lines 146-148 were moved to Materials and method section.
- All of a sudden, the authors mention ANOVA in line 150 although it was not mentioned in the abstract.
Ans:
Thank you very much. The statement was rewriten by following the reviewer’s comment.
- Lines 156-157 repetitive, please rephrase.
Ans:
Thank you very much. The statement was rewriten by following the reviewer’s comment.
- I see no scientific discussion at all, just describing the statistical results of the tests.
Ans:
Thank you very much. The discussion of the obtained results was revised and expanded in the revised manuscript.
- Lines 205-208 so what is the thermal conductivity of the two nano particles, mention it to support your claims.
Ans:
Thank you very much. The statement was cited to support the claim in the revised manuscript.
- Lines 208-210 same as above.
Ans:
Thank you very much. The statement was cited to support the claim in the revised manuscript.
- Lines 213-216 contradicting explanation with previous paragraph, before you said hardness of particles is good to reduce roughness, then you said those particles if too many they scratch the machined surfaces and increase roughness, so where is the break point? These are two contradicting results.
Ans:
Thank you very much for your valuable comments. The discussion for these statements was expanded, rewriten, and cited to support the claim in the revised manuscript.
- Lines 221-228 does not read well at all, please rephrase and improve the justification. Also, did the authors measure cutting forces to confirm this claim?
Ans:
Thank you very much. The suggestion of the reviewer is very interesting. We did the cutting force measurement and analyzed the obtained data. The difference in the values of passive force and ratio of passive force/tangential force under MQL/MQCL conditions with/without nano cutting fluids. Therefore, the authors ask the reviewer's permission to cite in the revised manuscript.
Here are the images of cutting force measurement. The tangential force is blue and the passive force is pink. But currently, the work is under investigating process and will be discussed in the next paper.
In the content of this article, the author focuses on studying the influence of the input parameters on surface roughness. The study on cutting forces, tool wear and tool life will be discussed in the next study.
- Lines 229-231, the authors again make false claim, they say that Ra increase sharply, but looking at the graphs in figure 4, Ra change is less than 0.2 microns! This is very low and should not be considered signficant at all especially when machining steels.
Ans:
Thank you very much. The statement was rewriten. In Figure 6,f, the change of Ra is about 0.4 microns. Although the change of Ra is very small, in finishing, only a very small improvement of the surface roughness value Ra also has the great practical significance, sometimes changing a level of surface smoothness.The results of this study will be applied to the finishing processing of the mechanical parts after heat treatment, which requires very high accuracy.
- Line 244 “brings out the better results than emulsion oil under MQL condition” this sentence does not read well at all.
Ans:
Thank you very much. The statement was rewriten in the revised manuscript.
- The results are merely described and is limited to comparing the experimental observation and describing results. The authors are encouraged to include a more detailed results and discussion section and critically discuss the observations from this investigation with existing literature.
Ans:
Thank you very much. The results and discussion were revised and expanded by adding more explanation and citing the relevant papers to support.
- Conclusion can be expanded or perhaps consider using bullet points (1-2 bullet points) from each of the subsections.
Ans:
Thank you very much. The conclusion section was revised and expanded by following the reviewer’ comments.
- After reading the paper i found the authors did not make any attempt to provide any comparison of their results with past studies similar or closely related to this work, there is no novelty or sceintific discussion in the discussion. therefore, paper can not be accepted at its current form and authors must extensively improve their article.
Ans:
Thank you very much again. Reviewers have read and commented in great detail. The author team appreciates your comments. The paper was revised and modifed by following the reviewers’ comments to improve the quality.

Reviewer 4 Report
Dear Authors,
Congratulations on your work, which is focused on a very interesting subject. However, as any other paper in its first version, there is room for improvement. Thus, I'm providing to you some comments and suggestions trying to help you in improving your paper: Please pay attentiton to the following:
- Abstract is well drawn, but some quantitative results are missing. Please bring some results from the Conclusions to the Abstract, becoming the paper more appealing after a first look.
- Short or no attention is paid to the machinability of the alloy used (90CrSi). Due to the content of Si, this alloy is particularly abrasive. This should deserve some attention in the Introduction, as well as previous studies on the machinability of this alloy, even under different conditions - dry machining, flood lubrication, etc.).
- The Introduction should be written in a more direct speech, referring the authors, the motivations and the results obtained in a short way, allowing for a better platform to discuss the values obtained after the presentation of your Results.
- Please point out the standard used to characterize the alloy used in machining tests. Did you test the alloy hardness? No standard deviation is pointed out. Thus, please completely characterize the alloy used, in terms of chemical composition, mechanical strength, and so on.
- The reference of the tool says a lot about the tool, but there are other important data about the tools which are very important to understand the machining phenomena. Thus, please completely characterize the tool geometry and tungsten carbide used in its production (grain size, etc.).
- Please use always a space between values and units (example: 40 kHz).
- The selection of the variables level is not clear. Why Soybean oil used about double of the cutting parameters used using emulsion)?
- Despite is is frequently used as only parameter to evaluate the surface roughness, Ra represents an average value, not an accurate value. Thus, you must complete the analysis with other roughness parameter, namely Rz, Rq, Rmax ou Abbott-Firestone profile, to caracterize the surface profile as robust or weak.
- Using just two levels, any mistake in terms of measurement can become a disaster. Thus, in terms of Results, this work is extremelly poor.
- No Discussion is presented. This is a severe lack you must correct, comparing your results with other results previously achieved using other alloys or other tools. Without this, the reader is lost, because cannot contextualize your results.
- More than a simple analysis of the results regarding each set of variables, it would be very useful for the reader to get information about the phenomena behind each effect. This is what brings value to your work and, without this, the relevance of the paper is very low.
- Conclusions are just a reminder about what has been obtained thorough the statistical analysis of the results. This show the weakness of the paper, which can be deeply improved.
- References need to be revised, because they present severe lacks (example: [16]).
Best wishes,
Kind regards.
Author Response
RESPONSES TO THE REVIEWER 4
We are very grateful for the reviews provided by the editors and each of the external reviewers of this manuscript. Please see below, our detailed response to comments.
Congratulations on your work, which is focused on a very interesting subject. However, as any other paper in its first version, there is room for improvement. Thus, I'm providing to you some comments and suggestions trying to help you in improving your paper: Please pay attentiton to the following:
- Abstract is well drawn, but some quantitative results are missing. Please bring some results from the Conclusions to the Abstract, becoming the paper more appealing after a first look.
Ans:
Thank you very much. The abstract and conclusion were revised and rewriten in the revised manuscript.
- Short or no attention is paid to the machinability of the alloy used (90CrSi). Due to the content of Si, this alloy is particularly abrasive. This should deserve some attention in the Introduction, as well as previous studies on the machinability of this alloy, even under different conditions - dry machining, flood lubrication, etc.).
Ans:
Thank you very much for your valuable comments. The introduction to the machinability of 90CrSi steel was added and expanded in the revised manuscript.
- The Introduction should be written in a more direct speech, referring the authors, the motivations and the results obtained in a short way, allowing for a better platform to discuss the values obtained after the presentation of your Results.
Ans:
Thank you very much. The introduction section was expanded and the high-quality revelant papers were added to formulate the problem. The changes were marked in red color.
- Please point out the standard used to characterize the alloy used in machining tests. Did you test the alloy hardness? No standard deviation is pointed out. Thus, please completely characterize the alloy used, in terms of chemical composition, mechanical strength, and so on.
Ans:
The chemical composition and mechanical properties of 90CrSi steel according to DIN 17350-80 standard were added in Tables 1, 2. The hardness of sample was measured by Mitutoyo HR-521
in the revised manuscript.
- The reference of the tool says a lot about the tool, but there are other important data about the tools which are very important to understand the machining phenomena. Thus, please completely characterize the tool geometry and tungsten carbide used in its production (grain size, etc.).
Ans:
Thank you very much for your valuable comments. The technical specification of the tool was added in Table 3 in the revised manuscript.
- Please use always a space between values and units (example: 40 kHz).
Ans:
The paper was revised carefully by following the reviewer’s comment.
- The selection of the variables level is not clear. Why Soybean oil used about double of the cutting parameters used using emulsion)?
Ans:
The selection of the variables level was based on the previous study, which was cited in the in the revised manuscript. Experimental design table was automatically exported from Design Expert 11 software (We don't know if we answered the question correctly).
- Despite is is frequently used as only parameter to evaluate the surface roughness, Ra represents an average value, not an accurate value. Thus, you must complete the analysis with other roughness parameter, namely Rz, Rq, Rmax ou Abbott-Firestone profile, to caracterize the surface profile as robust or weak.
Ans:
Thank you very much for your valuable comments. The aim of this paper is to survey the main effects of some input variable, and in further studies, we will perform the deeper study and supplement these parameters.
- Using just two levels, any mistake in terms of measurement can become a disaster. Thus, in terms of Results, this work is extremelly poor.
Ans:
Thank you very much. The content of this paper aims to evaluate the influence trend of the input factors, and on the basis of the results of this study, there will be orientation in choosing input parameters. and the value ranges of the parameters to optimize these parameters. These contents will be published in the next work.
- No Discussion is presented. This is a severe lack you must correct, comparing your results with other results previously achieved using other alloys or other tools. Without this, the reader is lost, because cannot contextualize your results.
Ans:
Thank you very much for your valuable comments. In this study, the authors aim to choose carbide inserts which are recommended for steels before heat treatment, and want to show the improment of machinability of carbide tools and enlarge for hardened 90CrSi alloy tool steel by using newly environmental friendly cooling and lubricating methods, namely MQL and MQCL using nanofluids. The obtained results indicate that this type of carbide inserts can be effecttively used for hard machining with the same cutting conditions with the same cutting conditions as those for un-heat treated steels. This will bring out the alternative solutions and economic benefits.
- More than a simple analysis of the results regarding each set of variables, it would be very useful for the reader to get information about the phenomena behind each effect. This is what brings value to your work and, without this, the relevance of the paper is very low.
Ans:
Thank you very much. The information about the observed phenomena was cited in the high-quality papers, which were added in the reference in the manuscript.
- Conclusions are just a reminder about what has been obtained thorough the statistical analysis of the results. This show the weakness of the paper, which can be deeply improved.
Ans:
Thank you very much. The conclusion was revised and rewriten to improve.
- References need to be revised, because they present severe lacks (example: [16]).
Ans:
Thank you very much. The reference was revised in the manuscript.

Round 2
Reviewer 2 Report
Accept in present form.
Author Response
RESPONSES TO THE REVIEWER 2
We are very grateful for the reviews provided by the editors and each of the external reviewers of this manuscript. Thank you very much again.
Reviewer 3 Report
Figure 1 add some text and arrows to show the readers what to look at in the image.
Abstract is still generic, i cant see any useful findings. please use numbers or % to clearly tell us what were your main findings!
Conclusion is same as the abstract, generic findings nothing specific about the work done and analysed outputs.
Paper still require English check.
Author Response
RESPONSES TO THE REVIEWER 3
We are very grateful for the reviews provided by the editors and each of the external reviewers of this manuscript. Please see below, our detailed response to comments.
1) Figure 1 add some text and arrows to show the readers what to look at in the image.
Ans:
Thank you very much. Some some text and arrows were added in Figure 1 in the revised manuscript.
2) Abstract is still generic, i cant see any useful findings. please use numbers or % to clearly tell us what were your main findings!
Ans:
Thank you very much. Abstract was revised and rewriten by following the reviewer’s comments. The highest material hardness for CNMG120404 TM T9125 carbide tools is 325÷350 HB (35 HRC), and now this insert type can be effectively used for cutting hardened steel with high hardness of 60÷62 HRC (increasing about 71.4÷77.1% of hardness) with the same cutting speed.The changes were marked in red color.
Grade : T9215 - Tungaloy Cutting Tools - Metal Working Tools
3) Conclusion is same as the abstract, generic findings nothing specific about the work done and analysed outputs.
Ans:
Thank you very much. Conclusion was revised and rewriten by following the reviewer’s comments. The changes were marked in red color.
4) Paper still require English check.
Ans:
Thank you very much. English language in this paper was carefully checked again. The changes were marked in red color.

Reviewer 4 Report
Dear Authors,
Following my initial opinion, this paper is too weak to be published. Although some improvements, the paper is far from the required quality usually presented by Fluids. You must pay attention to this sentence, which is just an example of the manuscript quality (taken from the Conclusions): "Al2O3 nanofluid reveals the better cooling and higher thermal conductivity than MoS2 nanofluid, so it contributes to reduce the surface roughness values." As you can see, the Authors are correlating phenomena without any support, i.e., a better roughness cannot be directly correlated to a better cooling effect neither to thermal conductivity. There are intermediate effects whose can promote the surface roughness variation. However, they are not described. This can be confusing to the reader, and it is not clear at all. However, this is just one example.
Author Response
RESPONSES TO THE REVIEWER 4
We are very grateful for the reviews provided by the editors and each of the external reviewers of this manuscript. Please see below, our detailed response to comments.
Following my initial opinion, this paper is too weak to be published. Although some improvements, the paper is far from the required quality usually presented by Fluids. You must pay attention to this sentence, which is just an example of the manuscript quality (taken from the Conclusions): "Al2O3 nanofluid reveals the better cooling and higher thermal conductivity than MoS2 nanofluid, so it contributes to reduce the surface roughness values." As you can see, the Authors are correlating phenomena without any support, i.e., a better roughness cannot be directly correlated to a better cooling effect neither to thermal conductivity. There are intermediate effects whose can promote the surface roughness variation. However, they are not described. This can be confusing to the reader, and it is not clear at all. However, this is just one example.
Answer:
The authors appreciate the comments of the reviewers. We have carefully reviewed the content of the article and the discussions, and we made appropriate adjustments according to the reviewer's comments. The changes were marked in red color in the revised manuscript.